# Solvent Extraction with Cyanex 923 to Remove Arsenic(V) from Solutions

**DOI:** 10.3390/molecules29020470

**Published:** 2024-01-17

**Authors:** Francisco Jose Alguacil, Esther Escudero, Jose Ignacio Robla

**Affiliations:** Centro Nacional de Investigaciones Metalurgicas (CSIC), Avda. Gregorio del Amo 8, 28040 Madrid, Spain; fjalgua@cenim.csic.es (F.J.A.); mebaquero@cenim.csic.es (E.E.)

**Keywords:** arsenic(V), Cyanex 923, extraction, stripping, acidic solutions

## Abstract

The removal of harmful arsenic(V) from aqueous solutions using Cyanex 923 (solvation extractant) was investigated using various experimental variables: equilibration time, the acidity of the aqueous phase, temperature, extractant and arsenic concentrations, and O/A ratio. Cyanex 923 extracted As(V) (and sulfuric acid) from acidic solutions; however, it could not be used to remove the metal from slightly acid or neutral solutions. The extraction of arsenic is exothermic and responded to the formation of H_3_AsO_4_·nL species in the organic phase (L represents the extractant, and the stoichiometric factor, n = 1 or 2, depends on the acidity of the aqueous phase). Extraction isotherms are derived from the experimental results. Both arsenic and sulfuric acid loaded onto the organic phase can be stripped with water, and stripping isotherms are also derived from the experimental results. The selectivity of the system against the presence of other metals (Cu(II), Ni(II), Bi(III), and Sb(III)) is investigated, and the ability of Cyanex 923 to extract As(V) and sulfuric acid compared to the use of other P=O-based solvation reagents, such dibutyl butylphosphonate (DBBP) and tri-butyl phosphate (TBP), is also investigated.

## 1. Introduction

Arsenic is considered one of the elements most harmful to humans and the environment; moreover, this element is labeled as one of the metals of murder [1]. Its presence in the environment is due to both natural and anthropogenic causes, as its toxicity depends upon the form in which it appears, either as gas or liquid effluents; whereas arsine gas is one of the most toxic gases, in liquid effluents, arsenic can occur as arsenic(III) and arsenic(V), and in both oxidation states as inorganic and organic species. The order of toxicity can be established as: inorganic As(III) > organic As(III) > inorganic As(V) > organic As(V).

Thus, the presence of these species in liquid effluents and drinking water is responsible for poisoning humans and life around the world, a problem that does not only exist in less developed countries, but also in industrialized ones [2]. It has been found that the presence of the above arsenic species in the organism causes different diseases, including cancer [3,4]. The maximum contaminant limit for arsenic in drinking water set by the EPA is 10 μg/L [5].

In addition to the above, arsenic is also present in metallurgical plants; probably the most noticeable example is the case of copper pyrometallurgical plants. Arsenic is one of the impurities normally found in copper concentrates and usually accompanies copper throughout the pyrometallurgical process (reverberation, conversion, and electrorefining operations), until its appearance in the copper electrolyte. At this stage, the increase in the arsenic concentration in the electrolyte results in the loss of electrolyte properties; thus, it is necessary to bleed it off to maintain the electrolyte’s properties. From this bleeding, arsenic is separated from sulfuric acid using solvent extraction and is recovered in the most suitable form.

From the above, we can infer that the removal of arsenic from various arsenic-bearing solutions is necessary, and in the particular case of arsenic(V), several technologies have been proposed to first isolate the element in a solution, and after, to recover it in the most suitable form.

Thus, some very recent investigations aimed to remove arsenic(V) from aqueous solutions through solvent extraction, ion exchange, adsorption and bioadsorption, etc. [6,7,8,9,10,11]. Though neglected by some authors, solvent extraction, if properly conducted, has advantages over other separation technologies, with its reasonable ease of scaling up, selectivity, very short operational time, and the possibility of the treatment of large volumes of contaminated water (solutions) in a short time.

This work presents an investigation into the removal of arsenic (V) from aqueous wastes using liquid–liquid extraction with Cyanex 923 extractant. Several variables influencing the extraction step are considered, as well as their relation to the stripping step. The extraction of arsenic is compared with the extraction of other metals and with the performance of other solvation extractants. Finally, the recovery of arsenic from strip solutions is proposed.

## 2. Results and Discussion

### 2.1. Arsenic(V) and Sulfuric Acid Extraction

In the first instance, the extraction of arsenic(V) is expected to be greatly dependent on the As(V) speciation in the aqueous phase as a function of the pH of this phase (Figure 1) [12]. Thus, as Cyanex 923 is a solvation reagent, extracting only neutral species, it is logical to suppose that the affinity of this reagent towards As(V) will only occur in the pH range in which the H_3_AsO_4_ species is predominant.

#### 2.1.1. Influence of the Acidity of the Aqueous Phase

The variations in arsenic(V) and sulfuric acid extraction with Cyanex 923 at various sulfuric acid concentrations in the aqueous phase were investigated next. In this case, the aqueous phase contained 1.5 g/L As(V) in the sulfuric acid medium (0.1–6 M), and the organic phases were 50% *v*/*v* Cyanex 923 in Solvesso 100. The results of these experiments are illustrated in Figure 2, showing that there was a continuous decrease in both the arsenic and sulfuric acid extractions as the initial sulfuric acid concentration in the aqueous phase increased. Table 1 shows the sulfuric acid to arsenic concentrations (molar scale) in the organic phase relationship at the various initial acid concentrations in the aqueous solution; these results show that this ratio was not constant, since it increased with the increase in the initial acid concentration in the solution. This lack of consistency should be indicative that the extractions of arsenic and sulfuric acid onto the organic phase were independent of one to another.

Other experimental results indicate that when using sulfuric acid concentrations below 0.1 M, the percentage of arsenic(V) extraction decreased, probably due to the increase in H_3_AsO_4_ deprotonation, and the formation of the non-extractable H_2_AsO_4_^−^ species in the aqueous phase (Figure 1). Thus, Cyanex 923 cannot be used to remove arsenic(V) from liquid effluents with a pH value near or greater than 2. In contrast to the above, Cyanex 923 was suitable for use to remove arsenic(V) from acidic solutions, i.e., copper refineries, though potential sources of arsenic-contaminated acidic solutions were not limited to these specific refineries [13].

#### 2.1.2. Influence of the Temperature

To establish the influence of the variation in temperature (20–70 °C) on arsenic(V) and sulfuric acid removal from solutions, a series of extraction experiments was carried out using an aqueous phase of 1.5 g/L As(V) in 3 M sulfuric acid medium, and organic phases of 50% *v*/*v* Cyanex 923 in Solvesso 100. The results from these experiments are shown in Figure 3, plotting the percentage of arsenic or sulfuric acid extracted onto the organic phase versus temperature. It can be seen that the percentage of arsenic extraction decreased as the temperature increased, whereas in the case of sulfuric acid this percentage of extraction remained almost constant.

The Vant Hoff equation established a relationship between the extraction constant (K_ext_) and the temperature; however, a series of investigations used an approximation to the above in which the relation of the temperature is established with the distribution coefficient of the given element [14,15,16,17,18], as follows:(1)$$\log D_{\mathrm{As}}=\frac{\Delta S^{0}}{2.3R}-\frac{\Delta H^{o}}{2.3\mathrm{RT}}$$
further, by plotting log D_As_ versus 1/T it is possible estimate the value of the change in enthalpy and entropy associated with the corresponding system. Moreover, it was described that this expression can be used in systems when the given component exists in the system as more than one species [19]. In the above equation, D_As_ (arsenic distribution coefficient) is defined as:(2)$$D_{\mathrm{As}}=\frac{{\left[\mathrm{As}\right]}_{\mathrm{org}}}{{\left[\mathrm{As}\right]}_{\mathrm{aq}}}$$
where [As]_org_ and [As]_aq_ are the arsenic concentrations in the organic and aqueous phases at equilibrium, respectively, and R is the gas constant. The results from this plot (r^2^ = 0.9644) indicate that arsenic extraction had an exothermic character (ΔH° = −19 kJ/mol), with ΔS° of −65 J/mol·K, indicating an increase in the order in the system as a consequence of arsenic loading onto the organic phase.

Since
(3)$$\Delta G^{0}=-{\mathrm{RTlnK}}_{\mathrm{ext}}$$
and taking into consideration the value of K_ext_ = 0.80 (see Table 3), it was found that the extraction process was not spontaneous.

#### 2.1.3. Influence of the Extractant Concentration

The influence of the variation in the extractant concentration on arsenic(V) and sulfuric acid extraction was also investigated. In these experiments, aqueous solutions contained 1.5 g/L As(V) and 1.5 M H_2_SO_4_, whereas the organic phases were of 25–75% *v*/*v* Cyanex 923 in Solvesso 100 and undiluted Cyanex 923 (previously saturated with water to avoid changes in the volume of the solutions); the possibility of the practical use of this undiluted extractant is an advantage over other reagents, since it allows one to use, if necessary, all of its chemical potential, and avoids the use of the organic diluent.

The results from these experiments are shown in Figure 4 (left). These results indicate that the percentage of arsenic and sulfuric acid extraction increased with the increase in the extractant concentration in the organic phase.

If the aqueous solution contained 1.5 g/L As(V) and 3 M H_2_SO_4_, the tendency found (Figure 4 (right)) was the same as in the above case; however, as was expected given the results shown in Figure 2, the percentage of extraction for both arsenic and sulfuric acid was lower than in the case of using the more highly diluted sulfuric acid solution.

Table 2 summarizes the [H_2_SO_4_]_org_/[As]_org_ molar relationship for the results shown in Figure 4. The results indicate that, at the various extractant concentrations, there was no direct relationship between these molar concentrations in the organic phase; thus, again, the extractions of both solutes by Cyanex 923 were independent of one another.

In order to define the species formed in the organic phase and its extraction constants, the experimental results were numerically treated using a tailored computer program that minimizes the U function, defined as:(4)$$U=\Sigma {\left(\log D_{\exp}-\log D_{\mathrm{cal}}\right)}^{2}$$
where D_exp_ and D_cal_ are the experimental distribution coefficients (Equation (2)) and the coefficients calculated by the program, respectively. In these calculations, the extraction of sulfuric acid by Cyanex 923 was also considered [20], whereas the formation of Cyanex 923 aggregates in the organic phase was not considered [21], but it can be generally stated that the formation of aggregates is more possible in aliphatic diluents than in aromatic ones. The results of these calculations are summarized in Table 3. Accordingly, the degree of extraction of arsenic(V) by Cyanex 923 responded to the formation of two species: H_3_AsO_4_·L and H_3_AsO_4_·2L at 1.5 M sulfuric acid, and a single species (H_3_AsO_4_·L) at 3 M sulfuric acid. Sulfuric acid was extracted via the formation of the H_2_SO_4_·L species in the organic phase [20].

#### 2.1.4. Influence of the Initial Arsenic Concentration in the Aqueous Phase

Also, the influence of the variation in the arsenic(V) concentration in the aqueous phase on metal extraction was investigated. In this case, the organic solutions were as above, whereas the aqueous phase contained 5 g/L As(V) and 1.5 M sulfuric acid. The results of these experiments are shown in Table 4; the results show that the variation in the arsenic concentration had very little influence on the percentage of arsenic extraction, whereas the percentage of sulfuric acid extraction remained the same as in the values obtained when the aqueous phase was 1.5 g/L As(V) and 1.5 M sulfuric acid.

With respect to the variation in the relationship between the [H_2_SO_4_]_org_/[As]_org_ molar concentrations at the various extractant concentrations, the results in Table 5 show that, again, there was no direct relationship between both solutes, and that this ratio was also different from that resulting following the extractions carried out with the aqueous phase of 1.5 g/L As(V) and 1.5 M sulfuric acid (see Table 2).

The almost identical percentages of extraction, and thus distribution coefficient values, indicate that in the extraction of arsenic(V), there was no formation of polynuclear complexes in the organic phase. These results are in accordance with the stoichiometries of the arsenic(V) Cyanex 923 species shown in Table 3.

#### 2.1.5. Influence of the O/A Relationship

Next we investigated the influence of continuous variations in the O/A ratio on arsenic and sulfuric acid extraction using Cyanex 923. In these experiments, aqueous solutions containing 1.5 g/L As(V) and 1.5 M H_2_SO_4_, or 1.5 g/L As(V) and 3 M H_2_SO_4_, were contacted with organic phases of 50% *v*/*v* Cyanex 923 in Solvesso 100 at various (0.5–4) O/A ratios. The results derived from these experiments are summarized in Figure 5 (l.5 M sulfuric acid, left; 3 M sulfuric acid, right). Both figures show the same pattern of an increasing percentage of solute extraction as the O/A ratio increased, since higher O/A ratios relate to a greater quantity of extractant available for solute extraction. In consequence, the percentages of As and sulfuric acid extracted were higher; however, and in accordance with the previous results (Figure 2), the percentages of arsenic are sulfuric acid extraction were slightly lower in the case of 3 M sulfuric acid than when the 1.5 M sulfuric acid solution was used in the experiments.

With respect to the separation of arsenic from sulfuric acid, Table 6 shows the values of the arsenic–sulfuric acid separation factors, defined as:(5)$${\mathrm{SF}}_{\mathrm{As}/H_{2}{\mathrm{SO}}_{4}}=\frac{D_{\mathrm{As}}}{D_{H_{2}{\mathrm{SO}}_{4}}}$$
where D refers to the corresponding distribution coefficients.

The results presented in Table 6 indicate that arsenic can be most effectively separated from the acid at low O/A ratios, though the co-extraction of both solutes was also a possibility, since they can be separated conveniently in the stripping stage (see below).

Figure 6 shows the corresponding arsenic (left) and sulfuric acid (right) extraction isotherms for the present system at the two sulfuric acid concentrations investigated in this work.

### 2.2. Arsenic(V) and Sulfuric Acid Stripping

Previous experiments showed that water was an effective strippant both for arsenic(V) and sulfuric acid; thus, no other chemical was investigated in this stage. Also, previous experimentations have shown that, in this stage, equilibrium was reached within a few minutes of contact between the As/H_2_SO_4_-loaded organic phase and the stripping phase.

In the case of arsenic(V) the stripping equilibrium can be represented by the following overall reaction:(6)$$H_{3}\mathrm{As}O_{4}\cdot nL_{\mathrm{org}}\Leftrightarrow H_{\mathrm{aq}}^{+}+H_{2}\mathrm{As}O_{4_{\mathrm{aq}}}^{2-}+nL_{\mathrm{org}}$$

This shows that, in a water medium, the equilibrium was shifted to the right due to the formation of non-extractable anionic arsenic species. In the above equation, n = 1 or 2.

In the case of sulfuric acid, and due to the use of a water medium, the corresponding extraction equilibrium was shifted to the left:(7)$$H_{2}{\mathrm{SO}}_{4_{\mathrm{aq}}}+L_{\mathrm{org}}\Leftrightarrow H_{2}{\mathrm{SO}}_{4}\cdot L_{\mathrm{org}}$$

#### 2.2.1. Influence of the Temperature

The influence of this variable on the stripping of arsenic and sulfuric acid loaded onto the organic phase was investigated using a 50% *v*/*v* Cyanex 923 in a Solvesso 100 organic phase loaded with 0.75 g/L As(V) and 0.46 M sulfuric acid. The results from these experiments are shown in Figure 7, plotting the percentage of arsenic or sulfuric acid stripped versus the temperature; it can be seen that with both solutes, the increase in the temperature produced an increase in the percentage of stripping.

In the stripping stage, we defined the distribution coefficient as:(8)$$D_{\mathrm{st}}=\frac{{\left[\mathrm{solute}\right]}_{\mathrm{aq}}}{{\left[\mathrm{solute}\right]}_{\mathrm{org}}}$$

Thus, if D_st_ > 1, it means that the stripping stage favored the removal of the solute from the organic phase to the stripping solution. Table 7 summarizes the distribution coefficients for arsenic and sulfuric acid in the stripping operation at various temperatures.

Thus, and in accordance with the results represented in Figure 7, the increase in temperature was accompanied by an increase in the corresponding distribution coefficient values. In the same table, the values of the separation factors H_2_SO_4_/As at each temperature are given. The separation factor was here defined as the ratio of the sulfuric acid to arsenic distribution coefficients:(9)$${\mathrm{SF}}_{H2\mathrm{SO}4/\mathrm{As}}=\frac{D_{H2\mathrm{SO}4}}{D_{\mathrm{As}}}$$
where the D values are defined as in Equation (8). If SF > 1, the separation of sulfuric acid from arsenic was favored. Thus, from the results shown in Table 7, we see that in the first instance, 20 °C seemed to be the best temperature to accomplish this separation, since the value of SF was the greatest.

Using the same thermodynamic expressions as in the extraction stage, it was concluded that, in the case of arsenic, the stripping stage had an endothermic character (ΔH° = 28 kJ/mol), with ΔS° of 94 J/mol·K, indicating an increase in the randomness of the system. With respect to the stripping of sulfuric acid, the process was also endothermic (ΔH° = 19 kJ/mol), with ΔS° of 78 J/mol·K, indicative of an increase in the system’s randomness.

#### 2.2.2. Influence of the O/A Relationship

Several experiments were carried out with the same organic and stripping phases as above to investigate the influence of the O/A ratio on the percentage of arsenic and sulfuric acid stripping. The results of these experiments are shown in Figure 8.

Similar to the above, Table 8 summarizes the distribution coefficients and the separation factor values at these various O/A ratios. These results show that, in both cases, the distribution coefficient value decreased with the increase in the O/A ratio; with respect to the separation factor values, the greatest value was obtained when an O/A ratio of 0.5 was applied. Thus, sulfuric acid was most effectively separated from arsenic using this ratio, and after this, arsenic was stripped from the organic phase using greater O/A ratios.

Lastly, Figure 9 shows the corresponding stripping isotherms.

### 2.3. Selectivity of the System

It was previously mentioned that Cyanex 923 cannot be used to remove arsenic(V) from aqueous solutions with pH values near to or greater than 2; instead, this extractant can be effectively used to remove this harmful element from acidic solutions, and one clear situation of potential use is in the removal of arsenic(V) from spent copper electrorefining solutions. Thus, in the present work, organic phases of 50% *v*/*v* Cyanex 923 in Solvesso 100 were put into contact with aqueous solutions containing Cu(II), Bi(III), Sb(III) and Ni(II), these elements being representative of metals presented in these electrorefining solutions.

Table 9 and Table 10 summarize the results derived from these experiments at 1.5 M and 3 M H_2_SO_4_, respectively; it can be seen that only Sb(III) was appreciably extracted by Cyanex 923, as the percentage of extraction increased with the increase in the acidity of the aqueous phase. In all cases, the extraction of sulfuric acid was near constant, suggesting again that the extractions of this mineral acid and of the metal were independent of one another.

Based on the results derived from this investigation, a possible sequence of steps for the treatment of these copper electrolytes includes:(i)Treatment of the copper electrolyte to remove antimony and bismuth. Several procedures have been proposed for the removal of these elements [22,23,24,25,26,27,28,29,30];(ii)Coextraction of arsenic and sulfuric acid with Cyanex 923;(iii)Stripping of sulfuric acid with water, with the possibility of reuse of the acidic solution;(iv)Stripping of arsenic with water, and Cyanex 923 recycled to another extraction step;(v)Precipitation of arsenic in the form of a salt to be safely dumped, preferably in a crystalline form such as scorodite [31,32,33,34,35].

### 2.4. Comparison with Other Solvation Extractants

In order to compare the performance of Cyanex 923 against those of other solvation reagents containing the P=O donor group, several experiments were carried out using Cyanex 923 (phosphine oxide), dibutyl butylphosphonate (DBBP, phosphonic ester) and tri-butyl phosphate (TBP, phosphoric ester), all of which were used in the undiluted form. Thus, a previous presaturation step with water was undertaken in all the cases, and these organic phases were put into contact with an aqueous solution containing 1.5 g/L As(V) and 1.5 M H_2_SO_4_.

The results of these experiments (Table 11) indicate that when using Cyanex 923, the highest percentages of extraction, both for arsenic and sulfuric acid, were reached, and a general extraction tendency of Cyanex 923 > DBBP > TBP was derived. This sequence is attributable to the greater electron-donor capacity of Cyanex 923 (R_3_PO) with respect to DBBP (R′(RO)_2_PO) and TBP ((RO)_3_PO), representing R different alkyl chains.

## 3. Materials and Methods

### 3.1. Materials

Cyanex 923 was used as received from the vendor (Cytec Ind, now Solvay, France). Its composition is given elsewhere [36] as comprising four trialkyl phosphine oxides, but it is normally considered as one extractant composed of the average of four reagents, generally represented as R_3_PO, with R being the various alkyl chains associated to the phosphine group. Diluent Solvesso 100 (aromatic) was obtained from Exxon Chem, Iberia, Spain and was used as received from the manufacturer. All other chemicals were of AR grade.

### 3.2. Solvent Extraction Procedure

Liquid–liquid extraction experiments were carried out in thermostatted separatory funnels (100 mL) that enabled mechanical shaking via four-bladed glass impellers. In these funnels, aqueous and organic solutions were mixed at a 1/1 O/A (organic to aqueous) volume ratio (unless otherwise stated). After phase disengagement, arsenic was analyzed in the aqueous solution by ICP-OES (Agilent Technologies 5100, Spain), whereas the sulfuric acid loaded onto the organic phase was analyzed by direct titration of the corresponding phase with standard NaOH solutions, in ethanol medium, using bromothymol blue as the indicator. The percentages of arsenic and accompanying metals extracted into the organic phase were calculated according to:(10)$$\%\mathrm{Extraction}=\frac{[M{]}_{\mathrm{aq},0}-[M{]}_{\mathrm{aq},e}}{[M{]}_{\mathrm{aq},0}}\cdot 100$$
where [M]_aq,0_ and [M]_aq,e_ were the arsenic and metal concentrations in the initial and equilibrium aqueous solutions, respectively. The percentage of sulfuric acid extraction was calculated on the same basis. In the case of stripping experiments, the next equation was used to calculate the percentage of metals stripped:(11)$$\%\mathrm{Stripping}=\frac{{\left[M\right]}_{\mathrm{st},e}}{[M{]}_{\mathrm{org},0}}\cdot 100$$
where [M]_org,0_ represents the metal concentration in the metal-bearing organic phase feeding the stripping operation, and [M]_st,e_ represents the metal concentration in the stripping solution at the equilibrium. The percentage of sulfuric acid stripped was calculated on the same basis. Other concentrations were estimated from the mass balance.

Preliminary tests have demonstrated that the extraction of arsenic from acidic medium (pH < 2.2) reached equilibrium within five minutes of contact, independently of the extractant concentration in the organic phase or arsenic and sulfuric acid concentrations in the aqueous solution. Sulfuric acid extraction reached equilibrium at even shorter times, and it was demonstrated that Cyanex 923 extracted sulfuric acid from aqueous media [20,37,38,39,40].

## 4. Conclusions

The extraction of arsenic(V) using the Cyanex 923 extractant was investigated. The results show that optimal arsenic recovery was achieved with around 0.5–2 M sulfuric acid, and that the extraction of arsenic was accompanied by the extraction of sulfuric acid, though it was demonstrated along the investigation that both extraction processes were independent of one another. The ability of Cyanex 923 to extract arsenic(V) from aqueous solutions decreased at pH values of 2 or greater than 2, thus in practical use it is not possible to extract arsenic(V) from slightly acidic or neutral As(V)-contaminated waters. The increase in temperature (20–70 °C) negatively affected the percentage of arsenic loaded onto the organic phase, which demonstrated the exothermic character of the extraction process. Arsenic(V) was extracted onto the Cyanex 923 organic phase via the formation of H_3_AsO_4_·nL (n = 1 or 2) from 1.5 M sulfuric acid solutions, and of H_3_AsO_4_·L from 3 M sulfuric acid medium. Sulfuric acid was extracted via the formation of H_2_SO_4_·L species in the organic phase. Extraction isotherms were provided during the experimental study.

Both arsenic(V) and sulfuric acid can be stripped from loaded organic phases via the use of water as a strippant, and stripping isotherms were also derived from the experimental study. Based in the above results, the separation of sulfuric acid from arsenic was better accomplished in the strip operation; first, sulfuric acid was stripped at low O/A ratios, and secondly, arsenic can be recovered from the As-loaded organic phase.

The experimental results show that out of a series of metals, only antimony(III) was extracted by Cyanex 923 from acidic solutions, with the extraction increasing with the increase in the sulfuric acid concentration in the aqueous feed solution, though this extraction was also lower than that of arsenic(V).

Cyanex 923 was a stronger extractant of arsenic(V) and sulfuric acid than other solvation reagents, such as dibutyl butylphosphonate and tri-butylphosphate, the above being attributable to the higher electron-donor capacity of Cyanex 923 in comparison with those of dibutyl butylphosphonate and tributylphosphate.

A sequence of steps for the treatment of an acidic copper electrolyte was proposed.

## Figures and Tables

**Figure 1 molecules-29-00470-f001:**
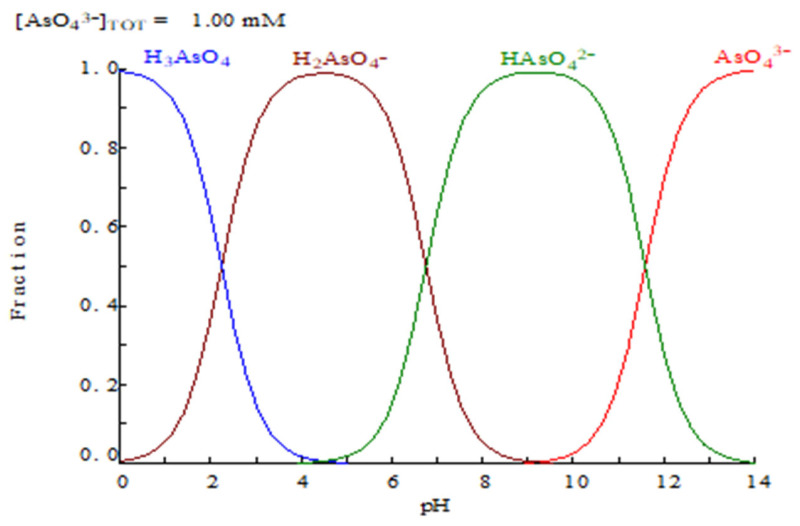
Distribution of arsenic (V) species versus pH [12].

**Figure 2 molecules-29-00470-f002:**
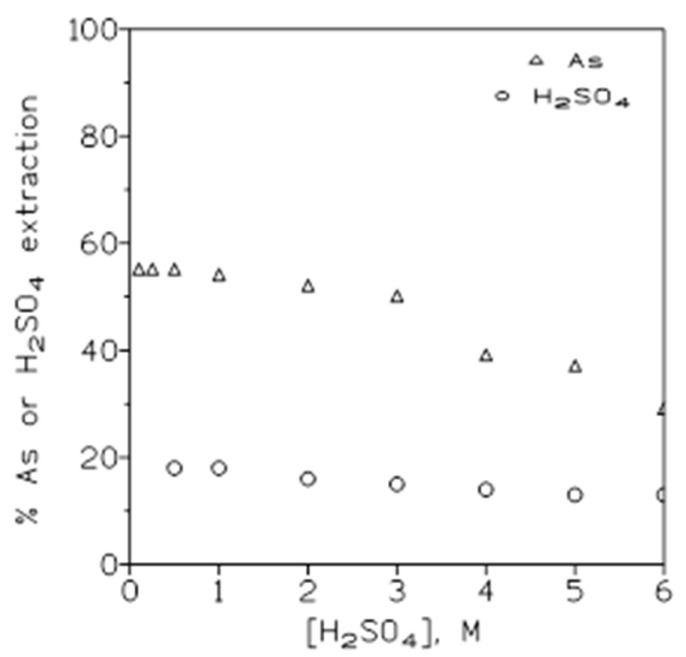
Variations in the percentages of arsenic and sulfuric acid extraction at various acid concentrations in the aqueous phase. Equilibration time: 5 min. Temperature: 20 °C. O/A ratio: 1.

**Figure 3 molecules-29-00470-f003:**
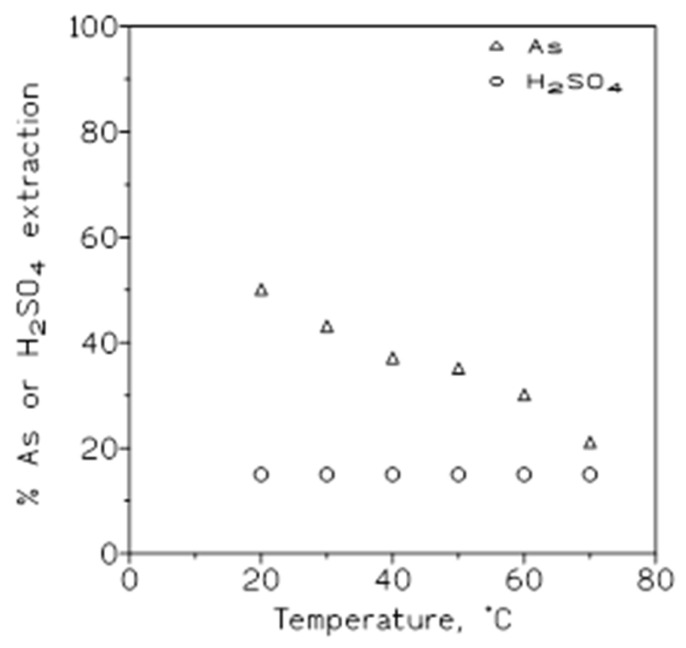
Arsenic and sulfuric acid concentrations at various temperatures. Aqueous phase: 1.5 g/L As(V) and 3 M H_2_SO_4_. Organic phase: 50% *v*/*v* Cyanex 923 in Solvesso 100. Equilibration time: 5 min. O/A ratio: 1.

**Figure 4 molecules-29-00470-f004:**
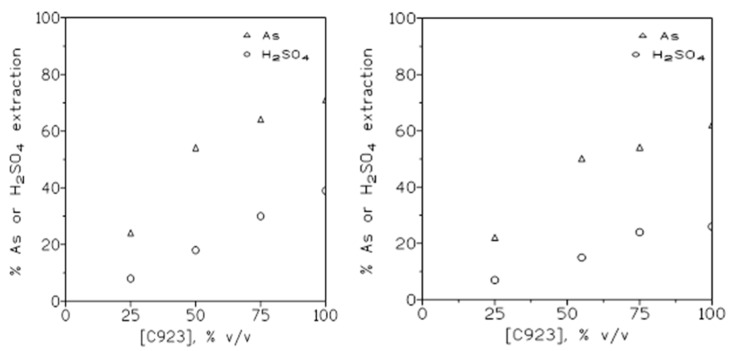
Arsenic and sulfuric acid extraction at various Cyanex 923 concentrations in the organic phase. (**left**): 1.5 g/L As(V) in 1.5 M sulfuric acid. (**right**): 1.5 g/L As(V) in 3 M sulfuric acid. Equilibration time: 5 min. Temperature: 20 °C. O/A ratio: 1.

**Figure 5 molecules-29-00470-f005:**
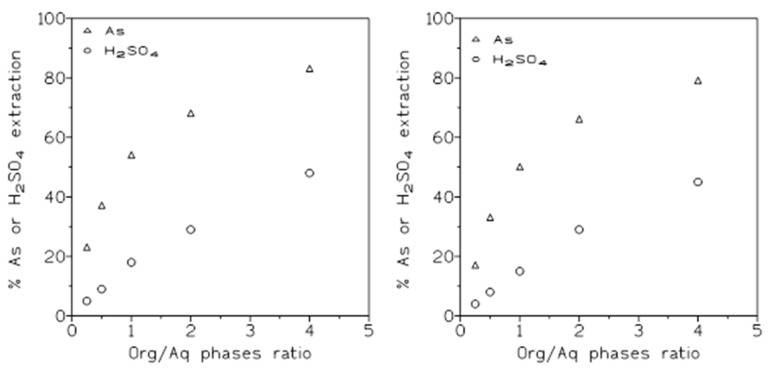
Arsenic and sulfuric acid extraction at various O/A ratios. (**left**): extractions carried out with 1.5 g/L As(V) and 1.5 M sulfuric acid. (**right**): extractions using 1.5 g/L As(V) and 3 M sulfuric acid. Equilibration time; 5 min. Temperature: 20 °C.

**Figure 6 molecules-29-00470-f006:**
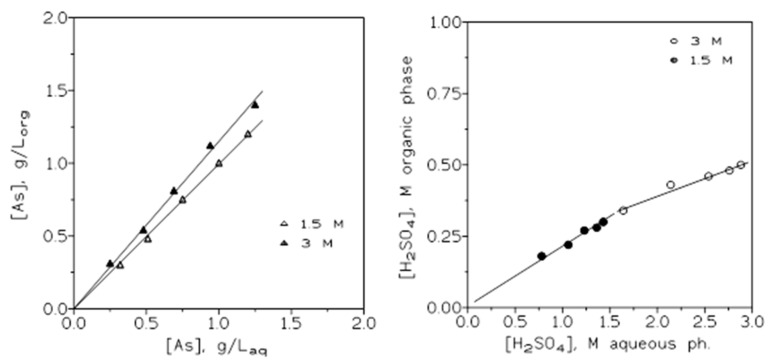
Arsenic (**left**) and sulfuric acid (**right**) extraction isotherms.

**Figure 7 molecules-29-00470-f007:**
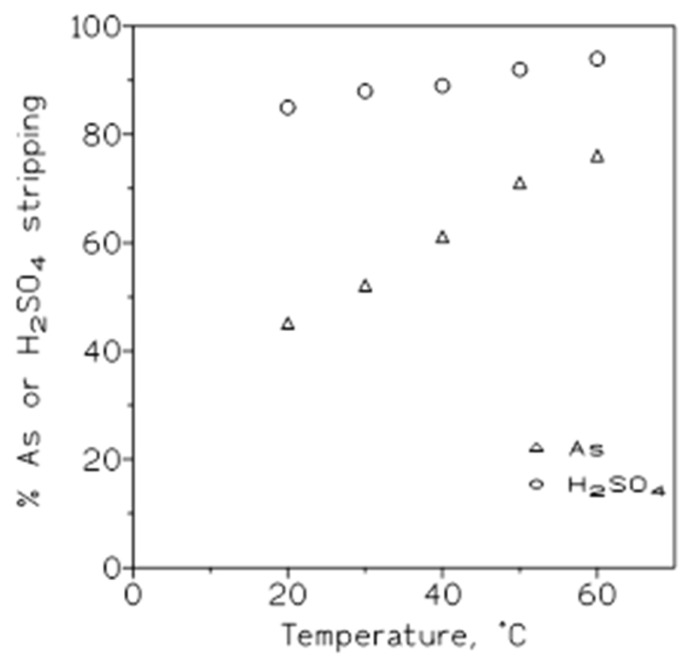
Plot of the percentage of arsenic and sulfuric acid stripping versus temperature. Stripping phase: water. Equilibration time: 5 min. O/A ratio: 1.

**Figure 8 molecules-29-00470-f008:**
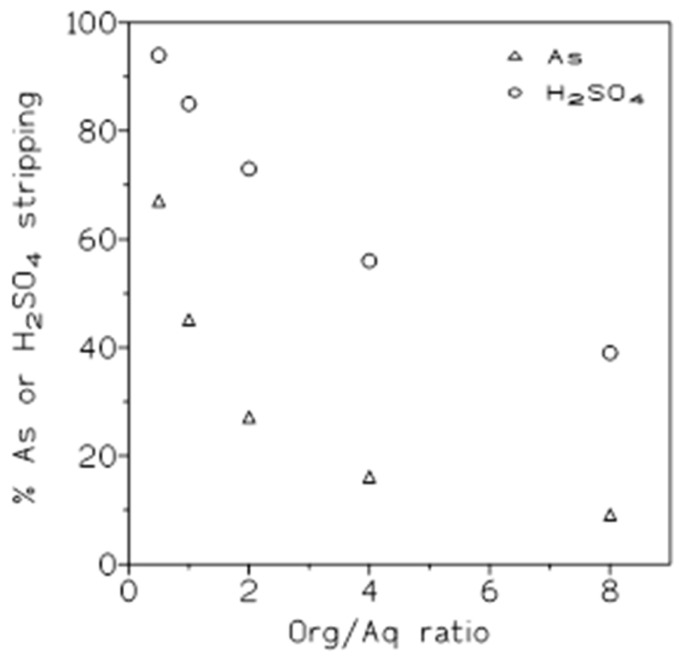
Percentages of arsenic and sulfuric acid stripped at various O/A ratios. Organic phase: 50% Cyanex 923 in Solvesso loaded with 0.75 g/L As(V) and 0.46 M sulfuric acid. Stripping phase: water. Temperature: 20 °C. Equilibration time: 5 min.

**Figure 9 molecules-29-00470-f009:**
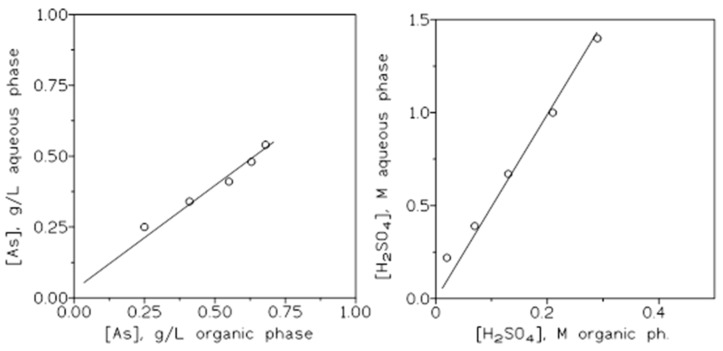
Arsenic (**left**) and sulfuric acid (**right**) stripping isotherms. Temperature: 20 °C. Equilibration time: 5 min.

**Table 1 molecules-29-00470-t001:** Variation in the [H_2_SO_4_]_org_/[As]_org_ relationship (molar scale) at various sulfuric acid concentrations in the aqueous phase.

[H_2_SO_4_], M	[H_2_SO_4_]_org_/[As]_org_
0.5	8
1	16
2	32
3	46
4	75
5	89
6	129

**Table 2 molecules-29-00470-t002:** Variation in the [H_2_SO_4_]_org_/[As]_org_ molar relationship at the various extractant and sulfuric acid concentrations.

Cyanex 923, % *v*/*v*	1.5 M H_2_SO_4_	3 M H_2_SO_4_
25	25	50
50	25	46
75	35	65
100	39	69

**Table 3 molecules-29-00470-t003:** Arsenic(V) species formed in the organic phase and extraction constants.

Sulfuric Acid, M	Species	K_ext_	U
1.5	H_3_AsO_4_·L	0.36	0.012
	H_3_AsO_4_·2L	0.67	
3	H_3_AsO_4_·L	0.80	0.011

L: Cyanex 923.

**Table 4 molecules-29-00470-t004:** Percentage of arsenic and sulfuric acid extraction at two initial metal concentrations in the aqueous phase.

Cyanex 923, % *v*/*v*	Arsenic	Sulfuric Acid
25	28 (24)	8 (8)
50	52 (54)	18 (18)
75	64 (64)	30 (30)
100	69 (71)	39 (39)

Equilibration time: 5 min. Temperature: 20 °C. O/A ratio: 1. Numbers in brackets corresponded to extractions with solutions of 1.5 g/L As(V) and 1.5 M H_2_SO_4._

**Table 5 molecules-29-00470-t005:** Variation in the [H_2_SO_4_]_org_/[As]_org_ molar relationship at the various extractant concentrations.

Cyanex 923, % *v*/*v*	[H_2_SO_4_]_org_/[As]_org_
25	6
50	8
75	10
100	13

**Table 6 molecules-29-00470-t006:** Values of the arsenic–sulfuric acid separation factors at the various O/A ratios.

O/A Ratio	1.5 M Sulfuric Acid	3 M Sulfuric Acid
0.25	5.74	5.55
0.5	5.78	5.55
1	5.32	5.55
2	5.46	4.70
4	5.39	4.48

**Table 7 molecules-29-00470-t007:** Variation in D_As_ and D_H2SO4_ with the temperature and the separation factors.

Temperature, °C	D_As_	D_H2SO4_	SF_H2SO4/As_
20	0.83	5.57	6.7
30	1.08	6.70	6.2
40	1.59	8.20	5.2
50	2.40	10.5	4.4
60	3.17	14.3	4.5

**Table 8 molecules-29-00470-t008:** Values of the distribution coefficients and separation factors at various O/A ratios.

O/A Ratio	D_As_	D_H2SO4_	β_H2SO4/As_
0.5	1	11	11
1	0.83	5.6	6.7
2	0.75	5.3	7.1
4	0.76	4.8	6.3
8	0.79	4.8	6.1

**Table 9 molecules-29-00470-t009:** Using Cyanex 923 to extract different metals at 1.5 M sulfuric acid.

Element	Concentration, g/L	%Extraction	%H_2_SO_4_ Extraction
Cu(II)	10	5	18
Bi(III)	1	nil	16
Sb(III)	0.07	26	17
Ni(II)	7	nil	17
As(V)	1.5	54	18

Equilibration time: 5 min. Temperature: 20 °C. O/A ratio: 1.

**Table 10 molecules-29-00470-t010:** Using Cyanex 923 to extract different metals at 3 M sulfuric acid.

Element	Concentration, g/L	%Extraction	%H_2_SO_4_ Extraction
Cu(II)	0.1	2	16
Bi(III)	0.1	nil	16
Sb(III)	0.07	25	16
Ni(II)	7	nil	16
As(V)	1.5	50	15

Experimental conditions as in Table 9.

**Table 11 molecules-29-00470-t011:** Arsenic(V) and sulfuric acid extraction using various solvation extractants.

Extractant	O/A Ratio	% As Extraction	% H_2_SO_4_ Extraction
TBP	1	18	2
	2	26	4
DBBP	1	46	10
	2	54	14
Cyanex 923	1	73	39
	2	88	50

Equilibration time: 10 min. Temperature: 20 °C. O/A ratio: 1.

## Data Availability

Data are contained within the article.

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
