# Peer review of "Solvent Extraction with Cyanex 923 to Remove Arsenic(V) from Solutions"

_molecules, 2024, doi:10.3390/molecules29020470_

Round 1
Reviewer 1 Report
Comments and Suggestions for Authors
The work is devoted to an important problem - the removal of arsenic (V) (a dangerous poison) from aqueous solutions, for example, from waste solutions of copper electrorefining. To solve this problem, the authors use an extraction method using the Cyanex 923 reagent in the Solvesso 100 aromatic diluent as an extractant. The authors examined the process of extraction of As(V) and sulfuric acid from different points of view. It was studied he influence of a number of factors on the extraction process and the percentage of extracted arsenic and sulfuric acid: pH, concentration of the extractant and extracted substances, temperature, etc. Extraction constants and thermodynamic characteristics of the process were determined. The process of distilling extracted components from the organic phase is also considered. The selectivity of extraction from aqueous solutions containing certain metals was studied, and Cyanex 923 was compared with a number of other extractants. The method proposed by authors can potentially be used for the removal of arsenic (V) from an acidic copper electrolyte. The work was carried out at a good experimental and theoretical level. However, there are a number of several comments:
1. It should be clarified, if the extraction percentage is a mass percentage and A/O ratio is the volume ratio?
2. It is advisable to write, how the Kext coefficients were determined.
3. Page 6. What initial data are used as the basis for calculating Dcal?
Despite these remarks, which are of a partial character, this work contains novelty, has practical value and can be published.
Author Response
Please attached file

Reviewer 2 Report
Comments and Suggestions for Authors
This article entitled “Solvent Extraction with Cyanex 923 to Remove Arsenic(V) from Solutions” deals with the removing of arsenic(V) from aqueous solutions by solvent extraction process. The authors purpose to study the extraction properties of the Cyanex 923, a phosphine oxide based extractant.
In this paper, a study of the extraction properties of As(V) is presented and is of great interest. The influence of different operating parameters as the acidity, the concentration of extractant or arsenic the temperature and the O/A volume ratio were studied. A selectivity study is presented using two different concentration of sulphuric acid. Finally, a comparison of Cyanex 923, TBP and DBBP is discussed for demonstrate the interest of Cyanex 923 in selective separation of As(V).
- The structure of the article and the writing is correct but should be improved in certain sentences.
- The scientific content is typical for a process study and developing. The content appears to be of quality and the presented research is of interest. But certain values (such as extraction factors, important when varying O/A in SX processes) and more in-depth interpretations are missing. For this type of article, the presentation and discution of distribution coefficient, extraction constant should be more adapted than %extraction (better for a thermodynamic interplaination).
However, in my opinion the article need to be improved before publication.
That's why I have some question/remarks in order to improve the quality of this article. The comments are my own and are mainly for constructive discussion.
General remarks
1/ The compositions of CYANEX 923 should be given and the general structure (it’s a phosphine oxides molecule !).
2/ The presentations of results in %X extracted is not the better. It is difficult to highlight the mechanism of extraction. I suggest that the authors plots D(X) or [X]org vs. variable, at equilibrium.
3/. Experimental methods: what is the experimental device separatory funnel was equipped with four blades? Is it possible to add a scheme to have a better understanding of the device?
Indeed, in general static extraction study are performed in tube and dynamic study in Mixer-settler devices.
4/ In order to avoid the presence of micro-droplets, are the phases centrifuged after separation? If so, it should be written.
Section 2.1.
5/ Be careful when crossing the line “Being Cyanex --- 923…”
6/ The quality of the figure 1 could be improved. Moreover, the solution conditions (temperature, compound used for variation of pH, others infomation etc……) used for the calculation should be specified.
Section 2.1.1.
7/ If authors not comment on the kinetic results, this section is not really needed. I suggest two possibilities: authors add kinetic results and comment the results OR they put this paragraph in part 3.2.
Section 2.1.2.
8/ The authors should develop the explanation. For a better understanding of the phenomena, the authors should not represents the %H2SO4 extracted. In my opinion, here the concentrations of acid in aqueous and organic phases increase.
I suggest that the authors re-plots points as : %As extraction OR D(As) vs. [H2SO4]aq at the equilibrium and [H2SO4]org vs. [H2SO4]aq at the equilibrium. These plotting should help authors and lecturer to highlights the influence of acidity.
Indeed, it is possible here that a competitive extraction between acid and As occu. Because the initial As/H2SO4 ratio vary from 0.04 up to 3 10-3 ([As]ini = 0,02 mol/L vs. [H2SO4]ini = 0,5 to 6 mol/L). In consequences, it is possible to suspect that the extraction of acids leads to a decrease of free extractant in organic phase and so a decrease of the extraction of As.
However, I am agree with authors that at low H+, the deprotonation of As species allows to a decrease of extraction.
Section 2.1.3.
9/ Same remarks, I suggest the authors shoud plot D(As) or [As]org vs. temperature and [H2SO4]org vs. temperature.
10/ The authors should specify why they use the approximation for the Van’Hoff equation (logD assimilated to logKex). This approximation is valuable by considering that only one stoichiometry is involved in the extraction mechanism. That’s right ?
For taking precautions, I suggest to tell about apparent enthalpy and enthropy.
Section 2.1.5.
11/ equation (5) change the beta by SF (separation factor) to avoid the confusion with a complexation constant.
12/ Results are logical, higher the O/A ratio is, higher the quantity of extractant is… In consequence : the amount of As and sulphuric acid extracted are higher. The authors should details their comments about results.
13/ Fig. 6 is not need if D(As) and D(H2SO4) are plotted in Fig 5.
Section 2.2.
14/ I am not really agreed with the use of Dst. A distribution coefficient is defined by the ratio D = [X]o/[X]a. When the D > 1, the L/L equilibrium is thermodynamically is rather towards solvent (and then extraction is favoured if O/A=1). When the D < 1, the L/L equilibrium is thermodynamically is rather towards aqueous (and then stripping is favoured if O/A=1).
I suggest the authors change the explaination. Moreover, the authors should be careful to not confuse the D as thermodynamic data and the extraction factor (defined by the D*O/A) which is an indication of performance of process.
Section 2.3.
15/ Table 9 is difficult to read. Two graphs ([H+] = 1,5 AND [H+] = 3 M) with istogramms as D(X) vs. element X at fixed acidity. Legend initial conditions.
Author Response
Please attached file

Round 2
Reviewer 2 Report
Comments and Suggestions for Authors
The article can be published in this form. The data being of quality and the work being clean.
However, thermodynamic data will remain difficult to read for thermodynamists.